# Effects of Evodiamine on Behavior and Hippocampal Neurons through Inhibition of Angiotensin-Converting Enzyme and Modulation of the Renin Angiotensin Pathway in a Mouse Model of Post-Traumatic Stress Disorder

**DOI:** 10.3390/nu16121957

**Published:** 2024-06-19

**Authors:** Zhixing Wang, Chengcai Lai, Baoying Shen, Bowei Li, Junru Chen, Xin Shen, Zhengping Huang, Chunqi Yang, Yue Gao

**Affiliations:** 1Medical College, Qinghai University, Xining 810016, China; wzx98748@163.com (Z.W.); asa2057516@163.com (C.L.); 2Department of Pharmaceutical Sciences, Beijing Institute of Radiation Medicine, Beijing 100850, China; shenby25511@163.com (B.S.); weiboli1029@163.com (B.L.); 15088094928@163.com (J.C.); shenxin9204@126.com (X.S.); ycq1qaz@outlook.com (C.Y.); 3Department of Neurology, Fujian Medical University, Quanzhou 362000, China; 15750848187@163.com

**Keywords:** post-traumatic stress disorder, evodiamine, renin angiotensin system, angiotensin converting enzyme, single prolonged stress

## Abstract

Post-traumatic stress disorder (PTSD) is a persistent psychiatric condition that arises following exposure to traumatic events such as warfare, natural disasters, or other catastrophic incidents, typically characterized by heightened anxiety, depressive symptoms, and cognitive dysfunction. In this study, animals subjected to single prolonged stress (SPS) were administered evodiamine (EVO) and compared to a positive control group receiving sertraline. The animals were then assessed for alterations in anxiety, depression, and cognitive function. Histological analysis was conducted to examine neuronal changes in the hippocampus. In order to predict the core targets and related mechanisms of evodiamine intervention in PTSD, network pharmacology was used. The metabolic markers pre- and post-drug administration were identified using nontargeted serum metabolomics techniques, and the intersecting Kyoto Encyclopedia of Genes and Genomes (KEGG) pathways were screened. Finally, the core targets were validated through molecular docking, enzyme-linked immunosorbent assays, and immunofluorescence staining to confirm the anti-PTSD effects and mechanisms of these targets. As well as improving cognitive impairment, evodiamine reversed anxiety- and depression-like behaviors. It also inhibited the reduction in the number of hippocampal neuronal cells and Nissl bodies in SPS mice inhibited angiotensin converting enzyme (ACE) levels in the hippocampus of SPS mice, and modulated the renin angiotensin pathway and its associated serum metabolites in brain tissue. Evodiamine shows promise as a potential candidate for alleviating the symptoms of post-traumatic stress disorder.

## 1. Introduction

Post-traumatic stress disorder (PTSD) is a mental illness triggered by trauma, characterized by intrusive reliving of past events, hyperarousal, avoidance behavior, and intense fear [1]. This chronic condition affects many individuals who show limited improvement with current therapies and is associated with an increased risk of suicide, psychological disorders, cardiovascular disease, and cognitive impairment [2]. Studies during the Corona Virus Disease 2019 (COVID-19) pandemic revealed a high estimated PTSD incidence rate of up to 17.52% across 24 countries [3]. Currently, only selective serotonin reuptake inhibitors (SSRIs) such as sertraline and paroxetine are approved for treating PTSD [4]. However, SSRIs have notable side effects, including initial neuroticism, emotional blunting, gastrointestinal issues, insomnia and sexual dysfunction, as well as drawbacks such as slow onset of action and uncertain effectiveness [5]. The development of new drugs and formulations for the treatment of PTSD is urgently needed.

Evodiamine (EVO) is a major alkaloid compound found in the dry unripened fruit Evodia fructus [6] that possesses a variety of pharmacological activities. With respect to the pharmacological effects of EVO, increasing attention has been given to its beneficial effects, including its anti-inflammatory [7], immune-modulatory [8], and antitumor effects [9], as well as its ability to retard the development of atherosclerosis [10]. Additionally, evodiamine has been found to be effective in treating cerebral ischemic injury [6] and chronic unpredictable mild stress-induced depression [11].

Although its mechanisms are not well understood, the renin angiotensin system (RAS) has been identified as a potential therapeutic target for PTSD [12]. Due to its involvement in sympathetic arousal, the renin angiotensin system has been directly linked to PTSD and cardiovascular disease (CVD) [13]. Renin is released by this hormone system, which regulates blood pressure. Stress activates the sympathetic nervous system, which releases renin into the bloodstream and activates angiotensinogen. The angiotensin converting enzyme (ACE) converts angiotensin I to angiotensin II, a peptide that increases blood pressure and vascular resistance. Additionally, angiotensin II inhibits norepinephrine reuptake and increases norepinephrine release, further enhancing sympathetic activity [14]. Therefore, given the well-established sympathetic overactivation in PTSD, the renin angiotensin system is a crucial target to better understand the increased CVD risk in PTSD patients [13].

Neuroprotective and cognitive protective effects of evodiamine have been demonstrated in an animal model of intracerebroventricular streptozotocin (ICV-STZ)-induced sporadic Alzheimer’s [15]. Despite its notable antidepressant-like activity and ability to improve cognition, the pharmacological potential of evodiamine has not been explored in a mouse model of PTSD. Therefore, molecular docking pharmacological network analysis was used to predict that evodiamine interferes with the renin angiotensin pathway through relevant targets for treating single prolonged stress (SPS)-induced PTSD. Through experimental validation, it was confirmed that EVO treatment attenuated the symptoms of SPS-induced PTSD in vivo, and the relevant bioactive constituents were identified through nontargeted serum metabolomics. Mechanistically, EVO interferes with the renin angiotensin pathway to alleviate PTSD mainly by inhibiting the angiotensin converting enzyme (ACE) in brain tissue, which may be a novel therapeutic target for PTSD.

This study investigated the histological and pharmacological alterations in the hippocampus of a mouse model of post-traumatic stress disorder (PTSD), along with the behavioral changes associated with PTSD. The study revealed that EVO exhibited anxiolytic and antidepressant properties, as well as improved memory and cognitive function in a post-stress behavioral task. Furthermore, the results demonstrated the neuroprotective effects of EVO on hippocampal neurons in stressed mice, including the prevention of hippocampal neuronal apoptosis. The renin angiotensin system (RAS) and other canonical systems have been documented as significant contributors to stress, particularly in the context of post-traumatic stress disorder. This paper provides additional insights into the involvement of ACE and RAS in stress-induced PTSD, as well as the potential for therapeutic intervention in stress and stress-related conditions.

## 2. Results

### 2.1. Effects of EVO on Behavioral Deficits in SPS-Induced PTSD Mice

The results of the Open Field Trial (OFT) showed that EVO and sertraline (SER) treatments significantly increased the percentage of distance in the center, total distance and average speed, and shortened incubation period in the center compared to the SPS group (Figure 1B–E). Moreover, in the elevated plus-maze test (EPMT), compared with SPS, EVO and SER led to significant increases in distance in the open arms and time spent in the open arms (Figure 1I,J). In mice, SPS-induced PTSD inhibits locomotor activity and spontaneous exploration, while EVO and SER reverse these effects. The results of the tail suspension test (TST) revealed that SPS notably prolonged the immobility time of the mice compared to that of the control group, whereas the EVO and SER treatments partially reduced the immobility time (Figure 1G). Additionally, compared with SPS, EVO and SER significantly decreased the immobilization time in the forced swim test (FST) (Figure 1H). These results suggest that EVO treatment has the potential to alleviate SPS-induced despair behavior in mice.

In order to test spatial learning, mice were trained in the Morris water maze (MWM), which entailed three trials per day for five consecutive days. The study revealed a significant decrease in escape latency across all groups during the training period. Particularly in the spatial navigation task, the latency to escape consistently decreased as the training days progressed for all groups of mice. This decrease was statistically significant in terms of the time and handling factors, with no significant interaction between these factors. Further analysis through a two-by-two comparison at different time points revealed that on day 4, the control group had a significantly shorter escape latency than the SPS group (Figure 1L). These findings suggest that mice in the control group effectively learned the task, while no notable difference in learning ability was observed among the experimental groups.

In the spatial exploration experiment, compared with those in the SPS group, the distance traveled to explore the target quadrant and the number of shuttles to the platform area in the EVO- and SER-treated mice were markedly greater (Figure 1M,N). These results indicate that SPS may mildly impair the memory capacity of mice, albeit without a significant effect, and that EVO has a substantial positive impact on this symptom.

The results demonstrated that administration of EVO significantly improved the depression-like behavior of SPS-exposed mice. Additionally, EVO treatment was expected to alleviate SPS-induced memory and cognitive deficits.

### 2.2. Effects of EVO on Hippocampal Morphology in SPS-Induced PTSD Mice

Morphological changes in the hippocampal region were assessed by HE staining (Figure 2A). Compared with the control group, the number of cells in the hippocampal CA1, CA2, CA3, and DG regions was significantly reduced in the model group (Figure 2B–E). The brain tissue structure was mildly abnormal, and a small number of neuronal cells in the hippocampal area in the field of view were degenerated with deepened staining, and no clear cytosol was seen (Figure 2A). On the contrary, compared with the model group, the hippocampal tissues in the EVO and sertraline-treated group tended to be normalized, including lighter staining, normal structure, more aligned, and more regular morphology with clear nuclei in the hippocampal CA1 region and DG region (Figure 2A), and a moderate dose of EVO significantly suppressed the reduction of neuronal cells in the hippocampus of the DG region in SPS mice (Figure 2E).

### 2.3. Effect of EVO on Nissl Bodies in the Hippocampus of PTSD Mice Induced by SPS

Nissl staining revealed that exposure to SPS resulted in a decrease in the number of Nissl bodies. However, the administration of EVO significantly ameliorated this effect (Figure 3). Specifically, in the hippocampal CA2 region, the Nissl bodies exhibited less pigmentation and a significant reduction in quantity in the model group than in the control group. Conversely, in the group treated with EVO, the coloration of neuronal Nissl bodies in the CA2 region of the hippocampus deepened, and the number of such vesicles increased. The results demonstrated that EVO could increase the number of Nissl bodies in the hippocampus of mice, providing a protective effect on the hippocampal neurons of mice with SPS-induced PTSD.

### 2.4. Network Pharmacology Reveals Potential Targets and Pathways Associated with Evodiamine Intervention in PTSD

To identify potential targets for treating PTSD, various disease target databases, such as GeneCards, OMIM, PharmGkb, TTD, and DrugBank, were consulted. A total of 2460 potential targets were identified from these databases. Further analysis using a Venn diagram identified 68 key targets that were common to both evodiamine and PTSD (Figure 4A). These targets, including ACE, REN, ADRB2, PTGS2, and INS, were identified as potential core targets of evodiamine in its therapeutic action on PTSD. The STRING database was used to construct a protein-protein interaction (PPI) network representing potential targets for evodiamine intervention in the treatment of PTSD (Figure 4B). The core of the PPI network was determined using the Cyto NCA plug-in, with filtering conditions set at values greater than the median. After two rounds of filtering, a total of five key targets were identified (ACE, REN, ADRB2, PTGS2, and INS), suggesting their significance in the therapeutic effects of EVO on PTSD (Figure 4C).

Gene Ontology (GO) functional enrichment analysis of the shared targets of evodiamine and PTSD revealed 1285 biological processes, 85 cellular components, and 103 molecular functions. Based on enrichment analysis results, evodiamine treatment for PTSD primarily affected tube diameter regulation, blood vessel diameter maintenance, and tube size control. Additionally, the molecular functions were primarily associated with synaptic membranes, postsynaptic membranes, and axon termini. In addition to neurotransmitter receptors, postsynaptic neurotransmitter receptors, and G protein-coupled amine receptors were all major components of the cell. These findings suggest that post-traumatic stress disorder (PTSD) involves various biological processes in living organisms and that evodiamine potentially exerts a therapeutic effect on PTSD by regulating these biological processes. The selection of the top 10 entries for the enrichment results bar and bubble plots was based on a significance level of *p* < 0.05 (Figure 4D).

Evodiamine and PTSD share 41 signaling pathways with a *p* value of 0.05, according to KEGG pathway enrichment analysis. The top 30 KEGG pathways, as determined through screening, were chosen for the creation of bar and bubble plots (Figure 4E). Among these pathways, renin secretion and the renin–angiotensin system were notably relevant to the core targets.

### 2.5. Effect of EVO on Serum Metabolites in SPS-Induced PTSD

Serum samples were collected from mice in the normal group, SPS model group, and EVO low-dose administration group for nontargeted metabolomic analysis based on the behavioral results. PLS-DA results show differences between the control group and the model group, as well as between the model group and the low-dose EVO treatment group (Figure 5A). This mode helps exclude irrelevant noise related to classification, thereby improving the analytical robustness and credibility of the model. To further identify potential biomarkers that play a significant role in metabolic differentiation, an OPLS-DA model was established for both the ESI+ and ESI− modes (Figure 5B). The parameters of the OPLS-DA replacement assay plot for the comparison between the OPLS-DA control and model groups in the ESI− mode were R2Y = 0.999 and Q2 = 0.256, while for the model group versus the EVO low-dose treatment group, the parameters were R2Y = 0.998 and Q2 = 0.497. These satisfactory parameters (R2Y and Q2) indicate reliable predictive ability for both aspects of the ESI model. In conclusion, the study findings validate the model constructed in negative ion mode. Significant differences were observed between the control and model groups and between the model and low-dose EVO treatment groups in negative ion mode. Moreover, EVO intervention significantly impacted the composition of serum metabolites in SPS mice. By applying thresholds of VIP > 1.0, multiplicity of difference (FC) > 1.5 or FC < 0.66, and *p* value < 0.05, the results of differential expression of metabolites in negative ion mode indicated the presence of one upregulated metabolite (Figure 5C). The model group exhibited one upregulated metabolite and two downregulated metabolites compared to those in the control group. Additionally, compared with the model group, the low-dose EVO treatment group displayed eight upregulated metabolites and twelve downregulated metabolites. Specifically, compared to the model group, the low-dose EVO treatment group exhibited eight upregulated and twelve downregulated metabolites. Notably, neuraminic acid was upregulated in the model group. Among the examined metabolites, neuraminic acid was upregulated in the model group and downregulated following EVO treatment. Cluster analysis of negative ion modeling for mid-differentially abundant metabolites in both the normal and model groups, as well as the administered and model groups, revealed that samples from the same group clustered together, suggesting a reasonably accurate screening for potential differentially abundant metabolites. (Figure 5D), enabling visualization of the contribution of metabolite differences across different subgroups. HMDB annotation and LIPID annotation analyses of differentially abundant metabolites in the negative ion model (Figure 5E) revealed the involvement of multiple small molecule metabolite pathways, including lipid and lipid-like molecules, organic acids and derivatives, organoheterocyclic compounds, benzenoids, phenylpropanoids, and polyketides. Furthermore, a variety of lipid structures and annotations were identified, such as steroids, bile acids and derivatives, steroid conjugates, sterols, secosteroids, and isoprenoids.

KEGG classification analysis revealed that the disease was associated with 45 metabolic pathway categories (Figure 5F). These pathways were primarily involved in the digestive system, endocrine system, nervous system, amino acid metabolism, and lipid metabolism. Further evaluation of the enrichment of KEGG-specific pathways revealed a total of 250 pathways. Among these pathways, central carbon metabolism in cancer, metabolic pathways, biosynthesis of amino acids, arachidonic acid metabolism, and biosynthesis of phenylpropanoids exhibited significant changes and were highly enriched. The results indicated that the metabolic marker neuraminic acid was upregulated in the model group and downregulated after EVO treatment. The KEGG classifications were primarily distributed across the digestive system, endocrine system, and nervous system.

### 2.6. Intersection of Network Pharmacology and LC-MS/MS KEGG Pathways

Through the intersection of 41 KEGG pathways acquired from network pharmacology with 250 KEGG pathways obtained from the negative ion mode of nontargeted metabolomics, a total of 24 pathways were identified (Figure 6A). Among these pathways, the renin secretion pathway was included. Furthermore, by considering the relationships between the potential targets identified from network pharmacology and the KEGG pathways, a more detailed analysis of the renin secretion pathway and the renin–angiotensin system was conducted, and the results were visually represented (Figure 6B). Additionally, the core targets within these pathway maps were identified and labeled accordingly (Figure 6C). The results demonstrated that EVO treatment for PTSD was associated with the renin secretion and renin–angiotensin system.

### 2.7. Molecular Docking Results for EVO and Potential Targets

Molecular docking was performed to determine whether EVO plays a significant role in regulating PTSD through relevant potential targets. The docking results show that EVO exhibits robust affinities for ACE, REN, ADRB2, and PTGS2, with binding energies of −8.5, −8.4, −8.5, and −9.7 kcal/mol, respectively. In ACE, the amino group of EVO forms hydrogen bonds with GLN-444 (Figure 7A). Similarly, in REN, the carbonyl group of EVO forms hydrogen bonds with TYR-83 (Figure 7B). In addition, the carbonyl group of EVO forms hydrogen bonds with THR-46 (Figure 7C). Moreover, the carbonyl group of EVO engages in hydrogen bonds with ARG-376, while its amino group forms hydrogen bonds with ASN-375 and GLY-225 (Figure 7D). After molecular docking, EVO exhibited a strong affinity for ACE, REN, ADRB2, and PTGS2.

### 2.8. Effects of EVO on Core Target Levels and the Renin–Angiotensin Pathway in the Hippocampus of SPS-Induced PTSD Mice

Network pharmacology is core target screening and prediction through databases. Molecular docking is to see whether EVO can be molecularly docked with the predicted targets. We initially identified the core targets corresponding to EVO using network pharmacology and molecular docking techniques. To further confirm these potential core targets, we performed ELISA experiments. The results showed that ACE levels were significantly elevated in the model group, which was reversed by EVO treatment, but the expression of REN, ADRB2, and PTGS2 remained unchanged (Figure 8A). To further confirm whether the changes in ACE expression were accurate, we further performed immunofluorescence experiments. The analysis demonstrated a significant elevation in ACE expression in the CA1, CA2, CA3, and DG regions of the hippocampus in SPS-treated mice, while EVO and SER treatments effectively counteracted these effects (Figure 8B,C). Finally, we found that ACE is indeed a core target, and EVO treats PTSD by affecting ACE expression and simultaneously affecting the RAS pathway.

## 3. Discussion

This study is the first to evaluate the therapeutic effect of EVO in a PTSD model in SPS mice. The single-prolonged stress (SPS) model, as introduced by Liberzon et al., represents a significant advancement in the study of post-traumatic stress disorder (PTSD) by successfully replicating alterations in the hypothalamic–pituitary–adrenal (HPA) axis observed in PTSD patients. This animal model incorporates three distinct stressors—2 h of restraint, forced swimming, and ether anesthesia—to induce psychological, physiological, and endocrine stress responses, respectively. The combination of these stressors results in a significant increase in serum corticosteroid levels, mimicking the symptom severity of PTSD in experimental settings [16]. Network pharmacology and nontargeted serum metabolomics verified that EVO has the potential to act as an anti-PTSD agent by reducing ACE levels in the hippocampus of SPS mice. This reduction subsequently modulates the renin–angiotensin pathway. PTSD leads to cognitive impairment, reduced recognition [17], and spatial memory impairment [18]. Various pharmacological activities of EVO have been reported, including antinociceptive and cognitive enhancement effects [19]. Further, evodiamine induces antidepressant-like activity by increasing the expression of the serotonin transporter (5-HTT) [20]. Recent studies examining the renin–angiotensin system in trauma-exposed individuals have offered additional evidence supporting its potential involvement in both PTSD and CVD. For instance, elevated circulating levels of renin have been observed in the trauma-exposed group, with the highest levels present in the PTSD group compared to healthy controls [21]. Inhibition of the renin–angiotensin system through the use of ACE inhibitors (ACE Is) and angiotensin receptor blockers (ARBs) has been linked to decreased anxiety and stress [22]. Experimental trials on isolated rat aortae revealed that EVO demonstrated endothelium-dependent vasodilatory effects [23]. Moreover, EVO reduces blood pressure by inhibiting aldosterone secretion via Ang II-related pathways [24]. However, the mechanism by which EVO exerts its anti-PTSD effects through ACE interference exhibits both similarities and differences compared to those of ACE inhibitors.

In this study, mice subjected to SPS exhibited prolonged time to OFT central zone motor latency, reduced central zone distance traveled (Figure 1B), and elevated anxiety indices in the EPM test (Figure 1G). The EPM test, a widely used paradigm, offers an unbiased assessment of anxiety-like behaviors in rodents, measured by the percentage of time spent entering or exploring the open arms [25]. Rodents displaying anxiety-like behaviors typically avoid entering the open arms in the EPM test, while anxiolytic medications typically increase their exploration of these open arms [26]. According to the findings of this study, SPS-exposed animals displayed a significant reduction in both the percentage of time spent in the EPM and the duration of time spent exploring the open arms, indicating an anxious state. However, EVO administration increased both the distance traveled and the time spent entering the open arms (Figure 1F), suggesting potential anxiolytic and antidepressant effects. The observed increase in immobility time in both the TST and FST reflects anxious behaviors and learned helplessness in the animals following trauma exposure (Figure 1D,E). EVO administration significantly decreased immobility time in both tests, indicating substantial improvement in depressive-like behavior among SPS-exposed mice, similar to the positive control sertraline.

Cognitive impairment and depression often cooccur in humans, and the relationship between these conditions is substantial [27]. Thus, in this SPS model, we evaluated the impact of EVO on cognitive performance using the MWM test. Both the hippocampus and amygdala are implicated in anxiety-like and depressive-like behaviors [28], with the hippocampus being particularly vital for learning and memory processes [29]. Animals rely on various maze cues to remember the location of hidden platforms in a pool, thereby completing the hippocampus-dependent spatial learning and memory challenge known as the MWM test [30]. Memory was quantified as the duration spent in the platform area during the test session after platform removal, while learning was measured by the decrease in time taken to locate the hidden platform across sessions [31]. The findings of this study indicate that prolonged exposure of mice to SPS stimuli in the MWM task led to heightened transfer latencies and reduced quadrant dwell times, demonstrating that EVO substantially mitigated hippocampus-dependent spatial learning and memory deficits (Figure 1H). Nonetheless, in the drug-treated group, escape latency consistently decreased with the progression of training days, exhibiting significant improvement by the fourth day (Figure 1H). Furthermore, mice treated with EVO showed a significant increase in the number of journeys and entries within the quadrant, a key measure of spatial memory retention (Figure 1H). Therefore, the findings of this study suggest that EVO treatment holds promise for mitigating SPS-induced memory and cognitive deficits.

The single prolonged stress (SPS) model is widely regarded as an appropriate animal model for PTSD and has been linked to hippocampal atrophy in confirmed patients [32]. Although PTSD is commonly understood as a prolonged, maladaptive stress response that does not directly affect the central nervous system [33], neuroimaging studies have revealed abnormal increases in activity within the cerebral cortex, hippocampus, and corpus callosum, along with alterations in gray matter volume and density in some PTSD patients [34]. Untargeted serum metabolomics results showed that the levels of neuraminic acid in serum were elevated in the model group compared to the normal group, and EVO treatment reduced neuraminic acid levels in the SPS model group. Studies have shown that neuraminic acid belongs to the family of salivary acids, which are based on neuraminic acid (Neu), which is usually found in the form of *N*-acetyl (NeuAc) or *N*-ethanolyl (NeuGc), or 2-keto-3-deoxynonanoic acid (Kdn) [35]. Additional studies have also shown that the term sialic acid (neuraminic acid) is used to refer to any member of this family [36]. Serum sialic acid (SSA) levels are abnormally high in pathological states exhibiting tissue destruction, tissue proliferation, or inflammation [37]. Whereas post-traumatic stress disorder (PTSD) is associated with increased inflammation. c-reactive protein (CRP) is a marker of systemic inflammation, and single nucleotide polymorphisms (SNPs) in the CRP gene have recently been associated with elevated blood CRP protein levels and increased disease severity in patients with PTSD [38]. There is also a significant correlation between serum salivary acid levels and cardiovascular disease (CVD) risk factors. In patients with CVD, elevated levels of SSA may be due to the release of free and bound sialic acid from damaged cardiomyocytes or vascular endothelium [39]. Since the renin–angiotensin system is involved in sympathetic arousal, it is directly associated with PTSD and CVD [13]. The EVO treatment group downregulated serum sialic acid and modulated the renin–angiotensin system, thus serum sialic acid has the potential to be a marker for the effectiveness of EVO in the treatment of PTSD, but further validation and research are needed. The hippocampus and prefrontal cortex, recipients of inputs from multiple sensory systems and higher cortical regions, are integral to the processes of learning and memory, while also playing crucial roles in regulating emotions and behaviors. Studies utilizing magnetic resonance imaging have demonstrated that individuals diagnosed with PTSD exhibit a reduction in hippocampal volume compared to unaffected controls, with reductions reaching up to 8%. Furthermore, the extent of this reduction is found to be directly correlated with the severity of PTSD symptoms [40]. In this paper, we found a significant reduction in the number of neuronal cells in the CA1, CA2, CA3, and DG regions of the hippocampus in the SPS model through pathological observation, mild abnormalities in brain organization, and a small number of neuronal cells in the hippocampus in the field of view were degenerated with deepened staining, and clear cytosolic bodies were not seen. While asymmetrical alterations in activity and augmented prefrontal cortex volume have been noted in certain studies involving PTSD patients, contrasting findings have documented cortical thinning and white matter atrophy in others [41]. Therefore, stress is thought to exert partial influence on the hippocampus and prefrontal cortex, implicating them in the pathogenesis of PTSD. Thus, the current study investigates behavioral changes and hippocampal neuron alterations in mice subjected to SPS [42]. Studies utilizing brain imaging techniques in PTSD patients have unveiled a reduction in hippocampal volume, a phenomenon that correlates with the severity of exposure to traumatic environments [43]. Moreover, mice exposed to SPS displayed a notable reduction in the Nissl bodies, while EVO treatment resulted in an elevation of Nissl bodies in the CA2 region of the hippocampus post-SPS exposure (Figure 3). Ultimately, hippocampal neuronal apoptosis is intricately linked to PTSD symptoms.

The core targets ACE, REN, ADRB2, PTGS2, and INS were screened by network pharmacology for the treatment of PTSD with EVO. ACE, REN, and ADRB2 were enriched in renin secretion and the renin–angiotensin pathway (Figure 6B). However, in addition to INS, we further investigated ACE, REN, ADRB2, and PTGS2. To validate the KEGG pathway identified through network pharmacology and assess EVO absorption into the bloodstream post-administration for its anti-PTSD effects in mice, we measured serum metabolites in EVO-treated mice from the normal, SPS-stimulated, and low-dose groups. We identified 43 differentially expressed metabolites that are frequently associated with PTSD-related pathways, including amino acid biosynthesis and the arachidonic acid metabolic pathway (Figure 5). Subsequently, by integrating the results of metabolomics and network pharmacology and comprehensively exploring the underlying mechanisms [44], we observed that both KEGG pathway analyses included the renin secretion pathway (Figure 6A). Additionally, serum metabolomics revealed greater endocrine correlations in lipid metabolism pathways in both the SPS model group and the EVO-treated group. In conclusion, EVO treatment for PTSD is associated with renin secretion and the renin–angiotensin system.

AT1R and ACE signaling hyperactivation exacerbates cognitive impairment, cell death, and inflammation in neurons [45]. AT1R and ACE signaling exacerbate oxidative stress in areas critical for cognitive function, such as the cortex, hippocampus, and basal ganglia. Following ischemic injury, activation of AT1Rs increases cholinergic and noncholinergic cell death in the rodent cortex and hippocampus [46]. Acetylcholine release from cholinergic neurons is decreased when ACE expression is upregulated [47]. Cholinergic cell death and dysfunction are frequently observed features of cognitive impairment [48]. Transgenic mice with three copies of the ACE gene were previously utilized to characterize the RAS system [49]. These transgenic mice exhibited high ACE activity, no change in blood pressure, and impaired short- and long-term memory, as measured by the new object recognition (NOR) test [50]. The molecular docking results demonstrated that EVO had a strong affinity for ACE, REN, ADRB2, and PTGS2 (Figure 6A). This finding was further verified by ELISA, which revealed that the hippocampal ACE level was significantly increased in SPS mice. Both the EVO group and the sertraline group showed a decrease in the ACE level. Similarly, immunofluorescence analysis revealed that the integrated optical density (IOD) of ACE in the CA1, CA3, and DG regions was greater in the SPS group than in the control group. However, EVO treatment reversed this phenomenon. These results suggest that EVO can modulate the renin–angiotensin pathway by downregulating hippocampal ACE levels in SPS mice and exerting a therapeutic effect on PTSD.

## 4. Materials and Methods

### 4.1. Animals

Healthy male C57BL/6 mice, weighing between 18 and 20 g, were obtained from Beijing Vitality River Laboratory Animal Technology Co., Ltd, Beijing, China (license number SCXK [Beijing] 2021-0006). These mice were individually housed in cages maintained at a temperature of 24–25 °C and a humidity level of 50 ± 5%, following a 12 h day-night cycle. Prior to the experiment, the mice were acclimatized for one week.

### 4.2. Stimulation of the SPS Model

With slight modifications, the SPS model was generated as previously described [51]. The SPS procedure comprised three stages: restraint (2 h), forced swimming (20 min), and ether anesthesia. Initially, mice were confined in a restraint tube for 2 h, ensuring unimpeded breathing through the smaller end of the cone. Subsequently, mice were subjected to forced swimming in a transparent acrylic cylinder (diameter: 240 mm, height: 500 mm) filled to two-thirds capacity with water at 24 °C, for a duration of 20 min. Following a 15 min respite, mice were anesthetized with diethyl ether until reaching a state of unconsciousness, after which they were returned to their respective cages.

### 4.3. Experimental Design

The mice were randomly allocated into six groups (*n* = 8): the control group, SPS stimulation model group, SER treatment group (15 mg/kg/day, i.p.), EVO low-dose group (5 mg/kg/day, i.p.), EVO medium-dose group (10 mg/kg/day, i.p.), and EVO high-dose group (20 mg/kg/day, i.p.). Sertraline (15 mg/kg, i.g.) was administered as a positive control, based on previous PTSD studies [52]. The three doses of EVO (high, medium, and low) were selected through pre-experimentation screening. At the conclusion of the dosing period, the mice were euthanized, and samples were collected for subsequent studies. Behavioral experiments were conducted on days 15, 16, 17, 18, and 19 (Figure 1A).

### 4.4. Behavioral Tests

#### 4.4.1. Open Field Test (OFT)

A well-established test for assessing mice’s locomotor activity, exploratory habits, and depression is the open field test [53]. A black square-bottomed open box, measuring 50 cm by 50 cm by 35 cm, supplied by Shanghai Xinruan Information Technology Co., Ltd., Shanghai, China, was utilized for the test. The mice were gently placed in the center of the square for a duration of 6 min. This period consisted of a 1 min acclimatization phase, followed by a 5 min observation phase. During the 5 min observation phase, the time spent by the mice in the center and their movement patterns were recorded. Subsequently, a 20% alcohol solution was employed to eliminate any lingering odors from the box after each test.

#### 4.4.2. Tail Suspension Test (TST)

The TST is a commonly used behavioral test to assess depressive-like behaviors in mice [54]. Using tape placed approximately 0.5 cm from the tail tip, each mouse’s tail was suspended from the edge of a shelf (Shanghai Xinruan Information Technology Co., Ltd., Shanghai, China). Video cameras were placed in front of the animals to track their movements while they were immobilized for five minutes. The TST experiment reflected anxious behavior and learned helplessness by immobility time.

#### 4.4.3. Elevated Plus Maze Test (EPMT)

The EPMT evaluates exploratory and anxiety behaviors in mice [55]. The maze consisted of two open arms, two closed arms, and a central area (dimensions: 35 cm × 5 cm × 10 cm, supplied by Shanghai Xinruan Information Technology Co., Ltd., Shanghai, China). Mice were positioned in the central area, facing an open arm, and permitted to explore for 5 min. Measurements included the distance traveled and frequency of entries into the open arm. Cleaning between sessions was conducted using a 20% ethanol solution. The EPM experiment reflects depressive behavior by detecting distance in the open arms and time spent in the open arms.

#### 4.4.4. Forced Swimming Test (FST)

The FST assesses the despair behavior of mice [56]. The mice were gently placed in a glass cylinder filled with water at 25 °C to a depth of 20 cm (Shanghai Xinruan Information Technology Co., Ltd., Shanghai, China). In the last 5 min of the 6 min test, the duration of immobilization was recorded. Software was used to analyze the movement of the mice using a video camera. The FST experiment reflected depressive behavior by immobility time.

#### 4.4.5. Morris Water Maze (MWM) Test

The MWM evaluates spatial learning and memory abilities [57]. It consisted of a circular pool with a diameter of 120 cm and a depth of 50 cm (Shanghai Xinruan Information Technology Co., Ltd., Shanghai, China). In the pool, a submerged platform (9 cm in diameter) was located. The water temperature was maintained at room temperature (25 ± 1 °C). Mouse swimming activities were monitored and recorded using a video analysis system. Each mouse underwent three training sessions per day with a 30 s rest between sessions. On days 1–5, the mice were released into the water from the two farthest quadrants from the platform. Navigating was allowed for 60 s for each mouse.

### 4.5. Histopathologic Analysis

Prior to HE staining, mouse brain tissues were fixed in 4% paraformaldehyde. The tissues were then dehydrated in varying concentrations of alcohol, embedded, and sectioned. Following dewaxing, the paraffin sections were stained with hematoxylin for nuclei and eosin for cytoplasm, dehydrated again, and sealed. The morphological changes in mouse brain tissue were assessed with an orthogonal optical microscope (Nikon Eclipse ci, Tokyo, Japan).

### 4.6. Nissl Staining and Neuron Counting

In order to evaluate neuronal damage in the hippocampal area, Nissl staining was performed. In brief, dewaxed hippocampal slices were stained using Nissl stain solution (toluidine blue method) according to the manufacturer’s instructions. The slices were then imaged using an orthogonal optical microscope (Nikon Eclipse ci, Tokyo, Japan) after dehydration, hyalinization, and mounting. Under a microscope, cellular morphological changes were observed in hippocampal CA1, CA2, CA3, and DG regions.

### 4.7. Network Pharmacological Analysis

#### 4.7.1. Collection and Screening of EVO Targets

Traditional Chinese Medicine System Pharmacology (TCMSP) database (https://tcmspw.com/tcmsp.php, accessed on 16 July 2023) was used to identify potential targets of EVO. During screening, the criteria used were an oral bioavailability (OB) ≥ 30% and a drug-likeness (DL) ≥ 0.18. Furthermore, the Bioinformatics Analysis Tool for Traditional Chinese Medicine Molecular Mechanisms (BATMAN-TCM) database (http://bionet.ncpsb.org/batman-tcm/, accessed on 16 July 2023) was used for screening, with a score of ≥20 and *p* ≤ 0.05 as the screening conditions. For further processing and analysis, detailed information on the relevant potential targets was compiled.

#### 4.7.2. Collection of Disease Targets

The disease targets were obtained through the GeneCards (https://www.genecards.org/, accessed on 16 July 2023), OMIM (https://www.omim.org/, accessed on 16 July 2023), PharmGkb (https://www.pharmgkb.org/, accessed on 16 July 2023), TTD (http://db.idrblab.net/ttd/, accessed on 16 July 2023), and DrugBank (https://www.drugbank.ca/, accessed on 16 July 2023) databases. The disease genes from the databases were merged using R language (R version 4.4.0), redundant gene entries were eliminated, and ultimately, PTSD-related genes were obtained.

#### 4.7.3. Constructing the PPI Network Diagram and Obtaining Core Targets

The gene symbols of the EVO and PTSD targets were uploaded to the STRING database (https://string-db.org/, accessed on 16 July 2023) to construct the protein interaction network. The species was specified as “Homo sapiens”, and the minimum interaction threshold was set to “highest confidence”. The network’s core was identified using the Cyto NCA plug-in by applying a filtering condition greater than the median value, leading to the identification of core targets after two rounds of filtering.

#### 4.7.4. Analysis of GO and KEGG Pathway Enrichment

The target proteins of EVO and PTSD were analyzed using R for GO function and KEGG pathway enrichment. GO function enrichment analysis encompassed BP, CC, and MF, with the top 10 entries selected based on *p* < 0.05 for visual analysis. KEGG pathway enrichment analysis identified the top 30 signaling pathways with *p* < 0.05 for visual analysis, shedding light on the mechanisms of pain and diarrhea treatment in PTSD. This analysis was crucial for understanding the biological functions and associated signaling pathways of EVO in treating PTSD.

### 4.8. LC-MS/MS-Based Serum Metabolomics Studies

After the behavioral analysis, eyeballs were extracted from each group of mice, and blood samples were obtained. Blood from the Con, Mod, and EVO_L (0.5 mg/kg/d) groups underwent centrifugation at 12,000 rpm for 15 min at 4 °C. Serum samples (50 μL) from these groups were then transferred to 1.5 mL EP tubes placed on ice, followed by the addition of 200 μL of cold methanol containing an internal standard. Following 2 min of vortexing, the samples were cooled for 10 min and subsequently centrifuged at 14,000× *g* for 15 min at 4 °C. The resulting supernatant (200 μL) was transferred to a new EP tube, then either lyophilized or concentrated via low-temperature centrifugation, and stored at −20 °C. Before online analysis, the concentrated metabolite extract was dissolved in 100 μL of 20% methanol/water solution until completely dissolved, and the supernatant was collected after centrifugation for analysis in both positive and negative ion modes.

### 4.9. Molecular Docking

The process of molecular docking is initiated with the retrieval of 2D structures of small molecule ligands from the PubChem database (https://pubchem.ncbi.nlm.nih.gov/, accessed on 17 July 2023). Converting these 2D structures into 3D formats followed using ChemOffice 14.0. Subsequent to this, the protein structures of the core targets were sourced from the PDB database (http://www.rcsb.org/, accessed on 17 July 2023) and visualized utilizing PyMol 2.2.0. Downloading the core target protein structures from the PDB database, the core protein was then isolated from the proto-ligand employing PyMol 2.4.0. Following isolation, the protein underwent hydrogenation, and Autodock 1.5.6 software was used to configure its atom types. Retrieving small molecule structures from the TCMSP database, they were saved as “mol2” files and subsequently employed in AutoDock and Vina for conducting the docking simulations.

### 4.10. Enzyme-Linked Immunosorbent Assay (ELISA)

ELISA kits for ACE, REN, ADRB2, and PTGS2 were procured from Jiangsu Enzyme Immunity located in Nanjing, China. The levels of these proteins in the hippocampal tissues of each mouse group were assessed using ELISA kits according to the manufacturer’s instructions. Optical density was measured at 450 nm with an enzyme-linked immunosorbent assay (ELISA) reader. Subsequently, the levels of ACE, REN, ADRB2, and PTGS2 in hippocampal tissues were determined based on the standard curve.

### 4.11. Immunofluorescence

For antigen retrieval, mouse brain tissue sections were subjected to immersion in a box containing EDTA antigen repair buffer (pH 8.0) within a microwave oven. Following this step, the sections were gently dried by shaking and a circular boundary was marked around the tissue using a histochemical pen to prevent antibody runoff. Next, an autofluorescence quencher was applied to the delineated circle for 5 min, followed by rinsing under running water for 10 min and subsequent incubation in serum for 30 min. Subsequently, rabbit anti-ACE (1:100, Proteintech, Wuhan, China) was added to the sections, which were then left to incubate overnight at 4 °C. Following this incubation, the sections underwent further incubation with a secondary antibody for 50 min at room temperature in the absence of light. Finally, the sections were stained with diaminophenylindole (DAPI) and observed under an orthogonal light microscope (Nikon Eclipse ci, Tokyo, Japan).

### 4.12. Statistical Analysis

The study’s data are expressed as mean values accompanied by their respective standard deviations. To compare data between two groups, a *t*-test was employed, while for comparisons among multiple groups, one-way analysis of variance (ANOVA) was utilized. Graphs were generated using GraphPad Prism 8.0.2 software. Statistical significance was defined as a *p* value below 0.05.

## 5. Conclusions

The present study pioneered the demonstration of the therapeutic effects of EVO in a PTSD model in SPS mice. Behavioral studies confirmed that EVO has anxiolytic and antidepressant effects and enhances memory and cognition in a behavioral task after SPS. The protective effect of EVO on hippocampal neurons and the attenuation of hippocampal neuron apoptosis in SPS mice were confirmed by HE staining and Nysted staining. In terms of mechanistic studies, mutual validation via network pharmacology and nontargeted serum metabolomics demonstrated that the improvements in these behavioral indices after SPS may be related to the modulatory effect of EVO on the renin angiotensin pathway. Additionally, using molecular docking, ELISA, and immunofluorescence staining, it was verified that EVO could regulate the renin angiotensin pathway by decreasing the ACE level in the hippocampus of SPS mice, thus exerting an anti-PTSD effect. This study is limited by the lack of clarification regarding the relationship between the upstream and downstream pathways of EVO in the treatment of PTSD. The specific mechanism by which EVO exerts its therapeutic effects through regulating the ACE-regulated RAS pathway has not been experimentally validated in both in vivo and in vitro settings, necessitating further research to comprehensively elucidate the mechanism of EVO’s anti-PTSD effects via the ACE and renin angiotensin pathways. Furthermore, the specific site or sites within hippocampal tissue or blood where EVO primarily exerts its effects remain to be elucidated, a critical consideration for the advancement of drug development (Figure 9).

## Figures and Tables

**Figure 1 nutrients-16-01957-f001:**
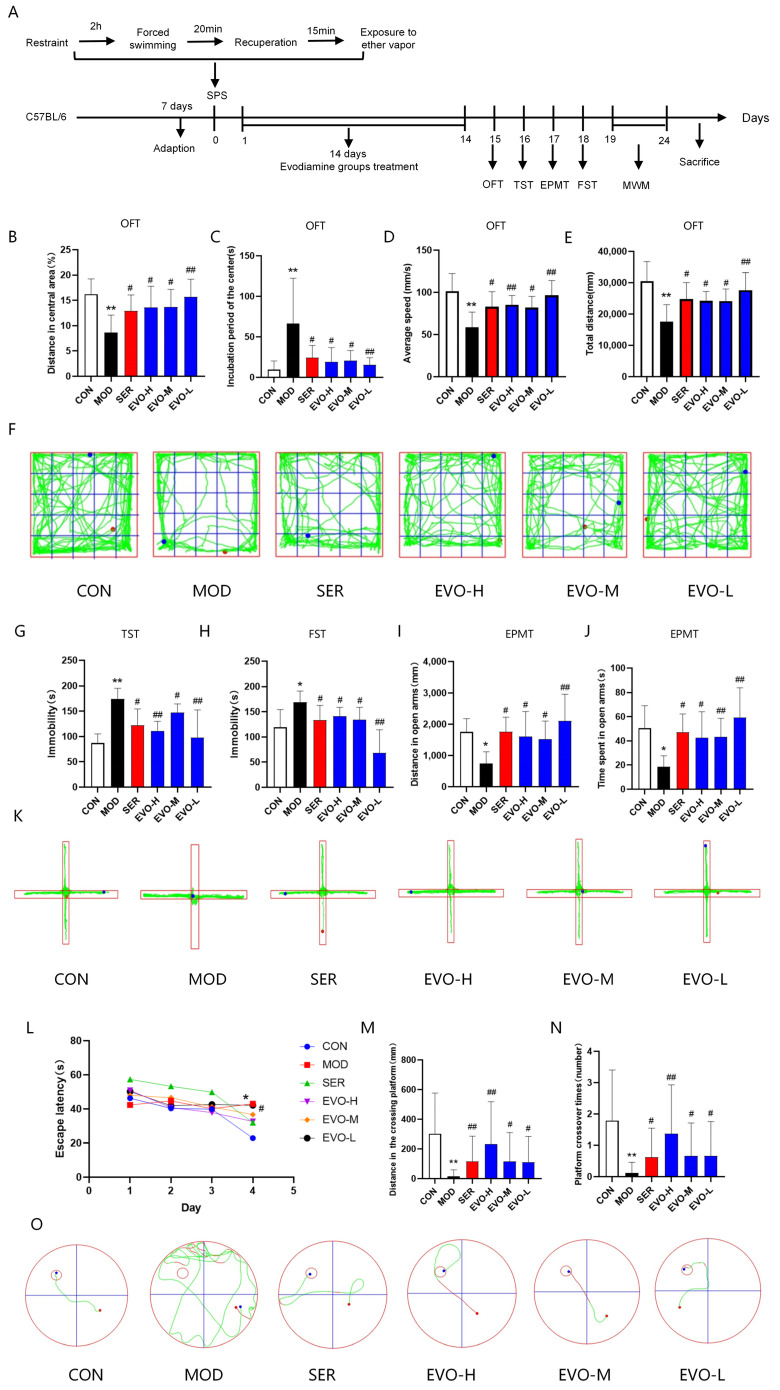
Effects of EVO on behavioral deficits in SPS-induced PTSD mice. (**A**) Experimental flow chart of the mouse model and treatment. (**B**) Percentage of distance in the center. (**C**) Incubation period in the center of the OFT. (**D**) Average speed in the OFT. (**E**) Total distance in the OFT. (**F**) Representative trajectory diagram of mice in the OFT. In the figure, the mice started at the red point and ended when they reached the blue point, while the green line indicates the trajectory the mice traveled. (**G**) Immobilization time in the TST. (**H**) Immobilization time in the FST. (**I**) Distance in the open arms in the EPMT. (**J**) Time spent in the open arms in the EPMT. (**K**) Representative trajectory diagram of mice in the EPMT. (**L**) Escape latency in the MWM test. (**M**) Distance in the crossing platform in the MWM test. (**N**) Number of platform crossover times in the MWM test. (**O**) Representative trajectory plots of mice in the MWM test. Forty-eight mice were randomly allocated into six groups: the control group (CON), model group (MOD), sertraline group (SER), evodiamine high dose group (EVO-H), evodiamine medium dose group (EVO-M), and evodiamine low dose group (EVO-L). The data are expressed as the means ± SDs (*n* = 8). * *p* < 0.05, ** *p* < 0.01 compared to the CON group; # *p* < 0.05, ## *p* < 0.01 compared to the SPS group.

**Figure 2 nutrients-16-01957-f002:**
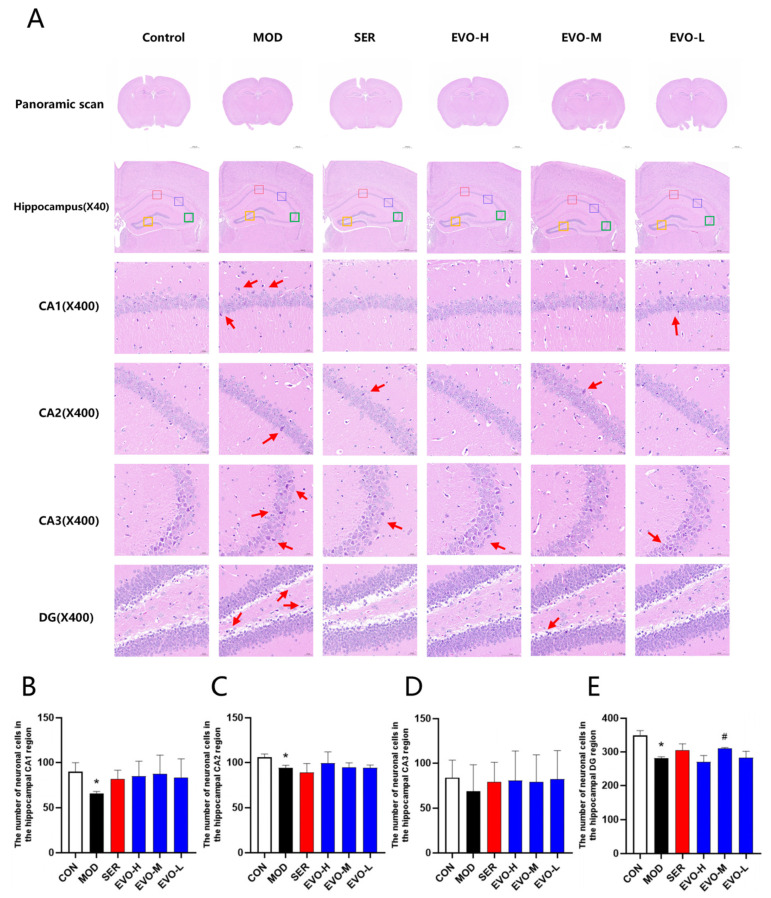
Effects of EVO on hippocampal morphology in SPS-induced PTSD mice. (**A**) The hematoxylin–eosin (H&E) staining was conducted on hippocampal tissue. Subsequent to staining, panoramic scanning was executed. The provided image displays HE staining of hippocampal regions CA1, CA2, CA3, and DG at a magnification of 400× (scale bar: 50 μm). The CA1 region is highlighted with a red box, CA2 with a blue box, CA3 with a green box, and the DG region with an orange box. (**B**) Quantification of neurons in the CA1 region of the hippocampus across each group of mice. (**C**) Evaluation of neurons in the CA2 region of the hippocampus within each group of mice. (**D**) Enumeration of neurons in the CA3 region of the hippocampus among all groups of mice. (**E**) Enumeration of neurons in the DG region of the hippocampus across each group of mice. Data are expressed as the means ± SDs (*n* = 3). Compared with the CON group, * *p* < 0.05; compared with the SPS group, # *p* < 0.05.

**Figure 3 nutrients-16-01957-f003:**
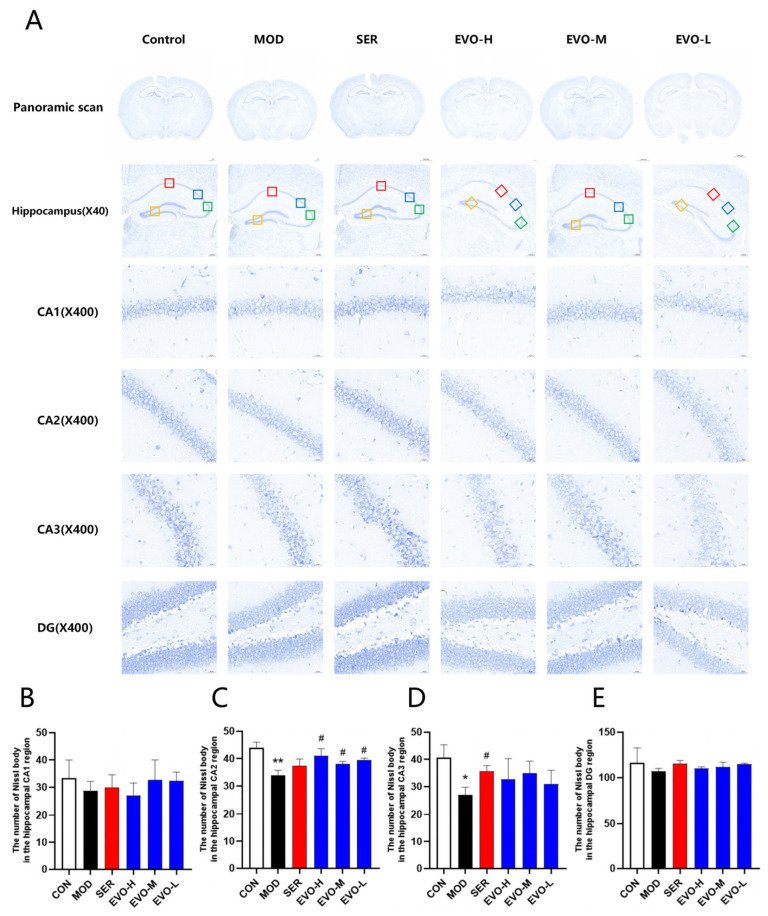
Effect of EVO on Nissl bodies in the hippocampus of PTSD mice induced by SPS. (**A**) Hippocampal Nissl staining representative images. Nissl staining images of the hippocampal CA1, CA2, CA3, and DG regions are shown in the upper panel, which is a panoramic scan of the brain tissue after Nissl staining, 400× magnification (scale bar is 50 μm). A red box represents the CA1 region, a blue box represents the CA2, a green box represents the CA3 region, and an orange box represents the DG region. (**B**) Quantification of Nissl bodies in the CA1 region of the mouse hippocampus across each group. (**C**) Assessment of Nissl bodies in the CA2 region of the mouse hippocampus within each group. (**D**) Enumeration of Nissl bodies in the CA3 region of the mouse hippocampus among all groups. (**E**) Enumeration of Nissl bodies in the DG region of the mouse hippocampus across each group. Data are expressed as the means ± SDs (*n* = 3). Compared with the CON group, * *p* < 0.05, ** *p* < 0.01; compared with the SPS group, # *p* < 0.05.

**Figure 4 nutrients-16-01957-f004:**
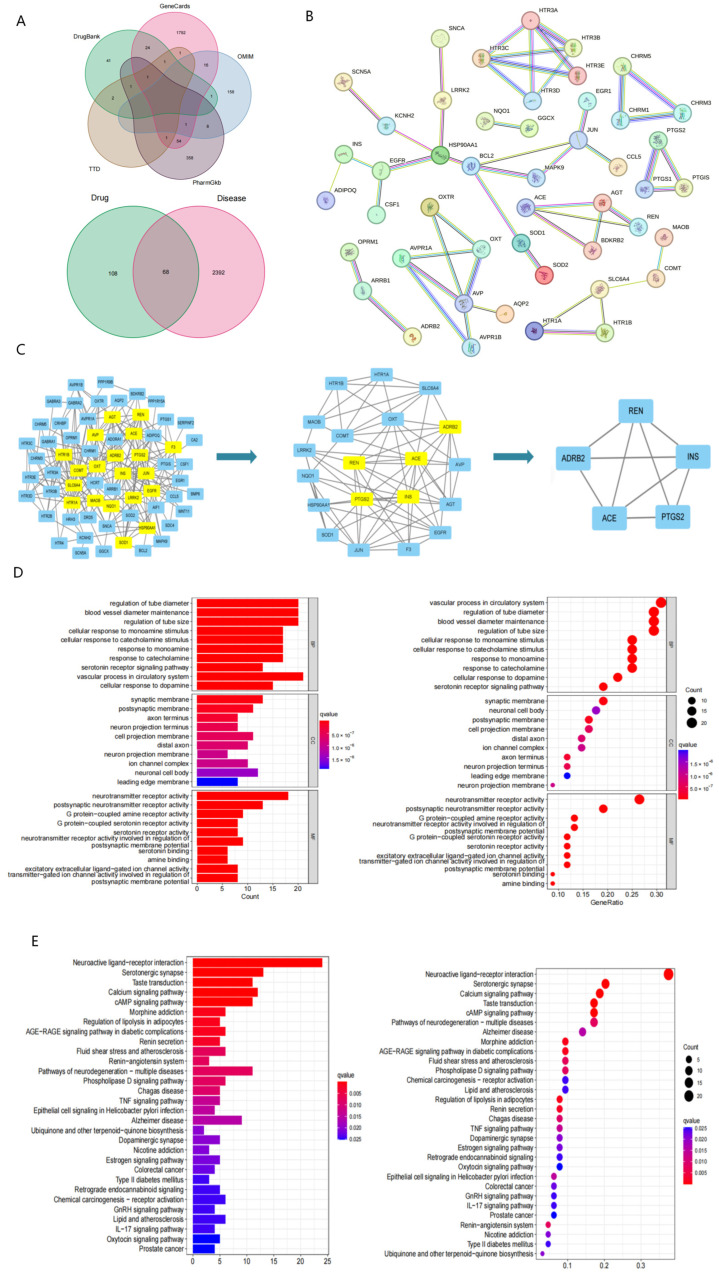
Network pharmacological analysis suggested potential anti-PTSD components in EVO. (**A**) Venn map of PTSD-associated genes combined with the Venn map and EVO-PTSD intersecting target genes. (**B**) Protein interaction network of EVO for PTSD. (**C**) Finding the core of the PPI network. (**D**) Results of GO enrichment plotted as bars and bubbles. (**E**) KEGG enrichment results in bar and bubble plots. The length of the bar and the size of the bubble indicate the number of targets enriched, and the color of the bar and node from red to blue indicates the *p* value from large to small. Therefore, the darker the red color is the greater the significance of the signaling pathway, indicating the greater importance of the signaling pathway.

**Figure 5 nutrients-16-01957-f005:**
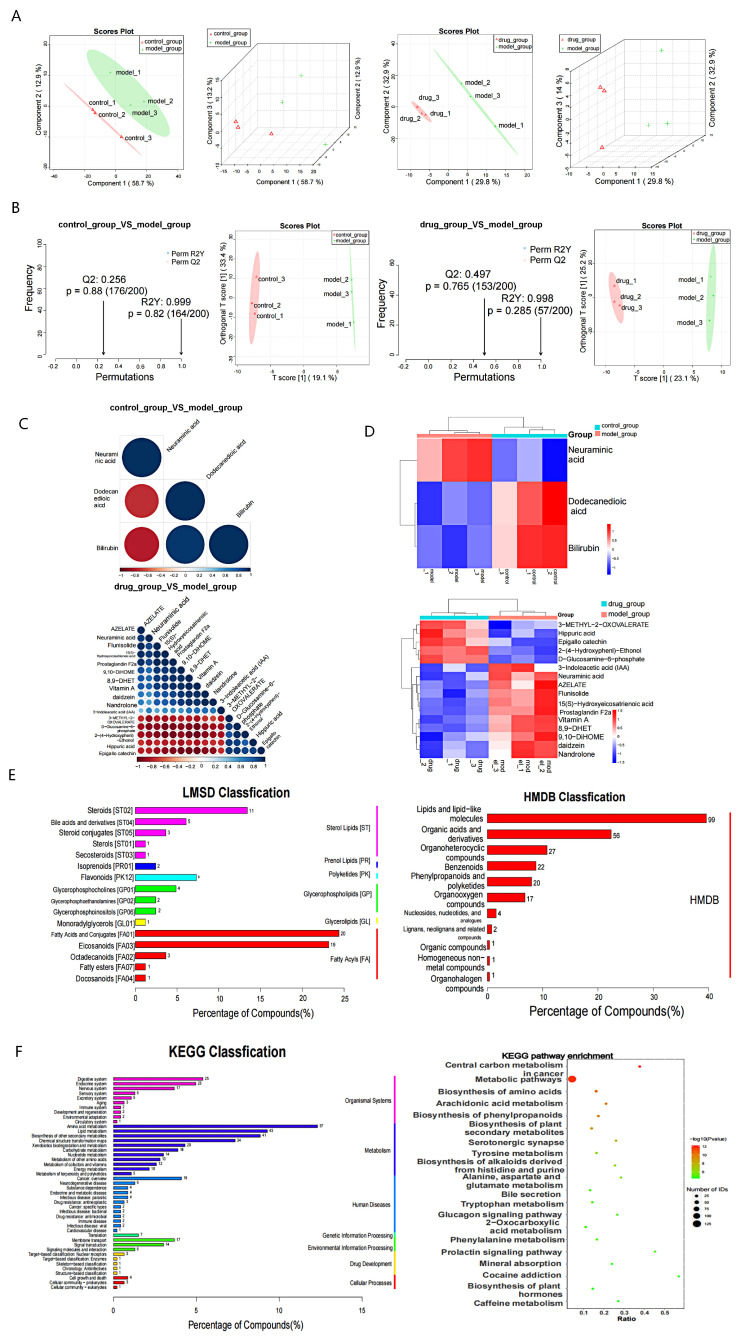
Nontargeted serum metabolomics analysis of the effects of EVO intervention on SPS-stimulated mice. (**A**) Score plots of PLS-DA for the normal and model groups, as well as the administered and model groups under negative ion mode. The horizontal axis represents the first principal component, the vertical axis represents the second principal component and the number following each principal component denotes the explained variance. (**B**) OPLS-DA substitution detection and score plots for the normal and model groups, and the administration and model groups under negative ion mode. R2Y and Q2 indicate the explained variance and predictability of the model, respectively. The horizontal axis of the plots represents the Spearman correlation coefficients between postdisruption and predisruption sample groupings. A correlation coefficient closer to 1 signifies a better fit. Typically, a Q2 value ≥ 0.2 is considered indicative of model reliability. (**C**) Correlation plots displaying differentially abundant metabolites between the normal and model groups, as well as the administered and model groups in negative ion mode. For positive correlation, the linear relationship tends toward 1 between two metabolites, whereas for negative correlation, it tends toward −1. Significance testing for metabolite correlation analysis was conducted, with a chosen threshold of *p* value < 0.05 indicating a significant correlation. (**D**) Cluster analysis plots were generated for comparing metabolites with significant differences between the normal group and the model group, and between the administration group and the model group, in negative ion mode. The horizontal axis represents sample numbers and their clustering relationship, while the vertical axis indicates the clustering relationship of metabolites. (**E**) Statistical plot of HMDB and LIPID statistics in negative ion mode. The horizontal axis displays the percentage of each metabolite type relative to the total number of entries, and the left vertical axis represents subclass entries. (**F**) KEGG statistical plot and pathway enrichment bubble plot in negative ion mode. The KEGG statistics plot illustrates the percentage of each metabolite type relative to the total number of entries on the horizontal axis, with subclasses (subclass) on the left vertical axis and major classes (class) on the right vertical axis. The KEGG pathway enrichment bubble plot presents the ratio on the horizontal axis, individual pathway names on the vertical axis, and colors representing the degree of enrichment (−log10(*p* value)), with circle size indicating the number of metabolic pathways.

**Figure 6 nutrients-16-01957-f006:**
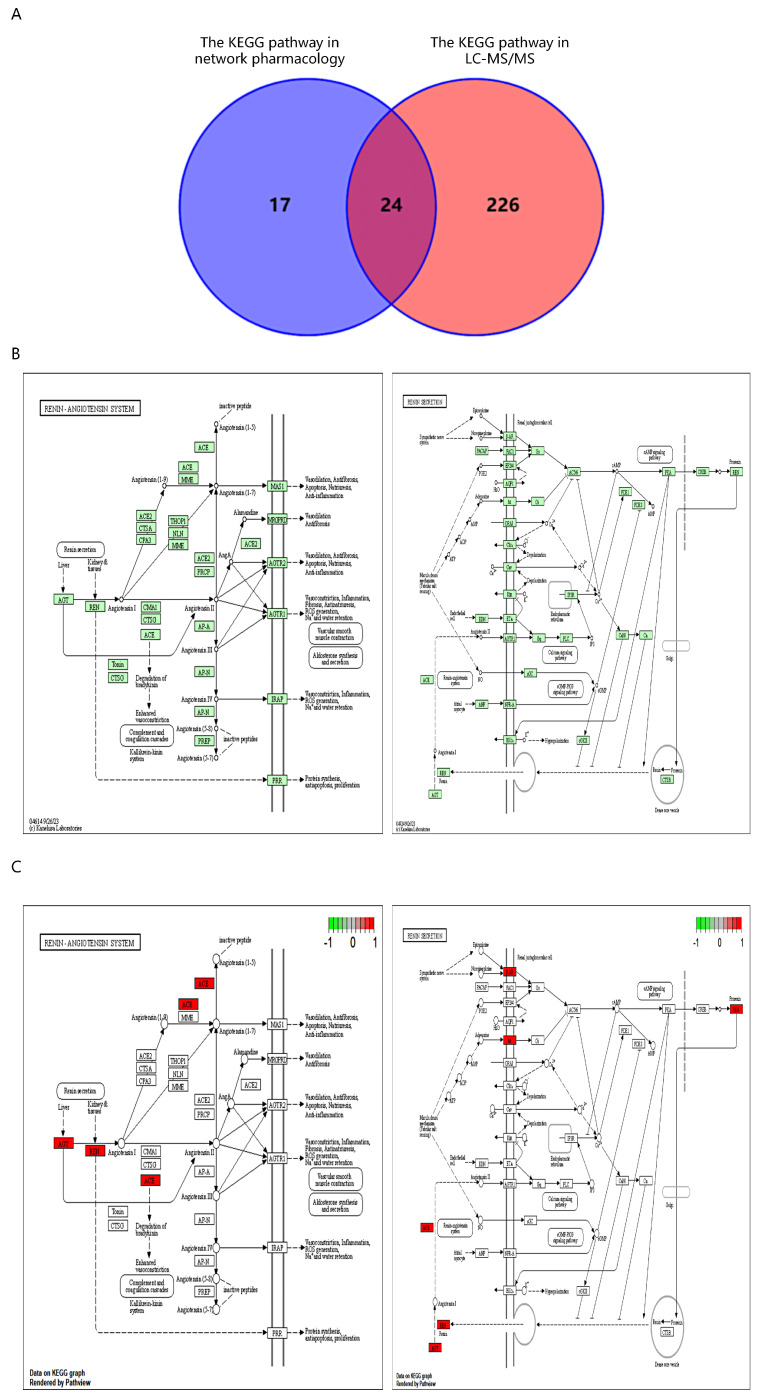
Network pharmacology and untargeted metabolomics of EVO intervention in PTSD associated with KEGG pathways. (**A**) Venn diagram of network pharmacology and off-target metabolomics intersecting KEGG pathways. KEGG pathways 1–24: Dopaminergic synapse, cAMP signaling pathway, morphine addiction, parathyroid hormone synthesis, secretion and action, taste transduction, regulation of lipolysis in adipocytes, longevity regulating pathway—multiple species, renin secretion, serotonergic synapse, neuroactive ligand–receptor interaction, endocrine resistance, Type II diabetes mellitus, GnRH signaling pathway, relaxin signaling pathway, estrogen signaling pathway, calcium signaling pathway, phospholipase D signaling pathway, oxytocin signaling pathway, cholinergic synapse, sphingolipid signaling pathway, ubiquinone and other terpenoid–quinone biosynthesis, retrograde endocannabinoid signaling, ovarian steroidogenesis. (**B**) The renin secretion and renin–angiotensin system pathway diagram. (**C**) Diagram of core target locations in the renin secretion and renin–angiotensin system pathways.

**Figure 7 nutrients-16-01957-f007:**
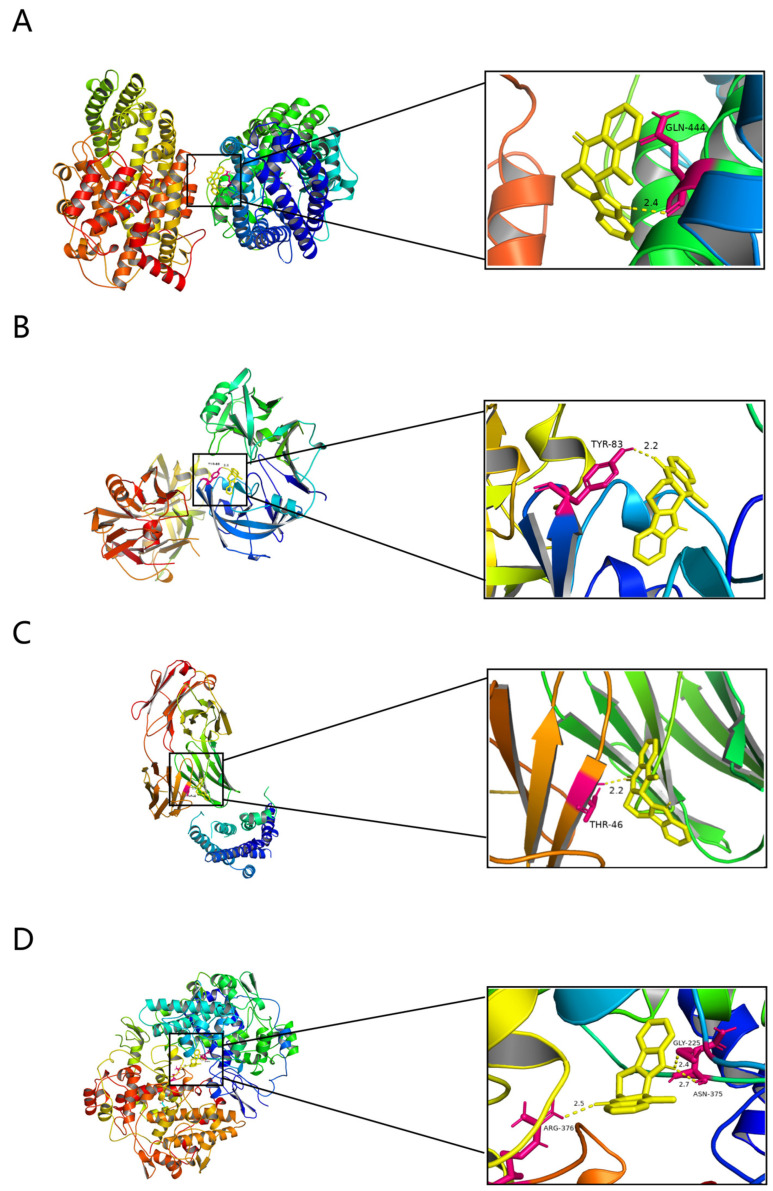
Molecular docking results. (**A**) A schematic diagram showing EVO’s binding mode and binding site details with ACE. (**B**) A schematic diagram showing EVO’s binding mode and binding site details with REN. (**C**) A schematic diagram showing EVO’s binding mode and binding site details with ADRB2. (**D**) A schematic diagram showing EVO’s binding mode and binding site details with PTGS2.

**Figure 8 nutrients-16-01957-f008:**
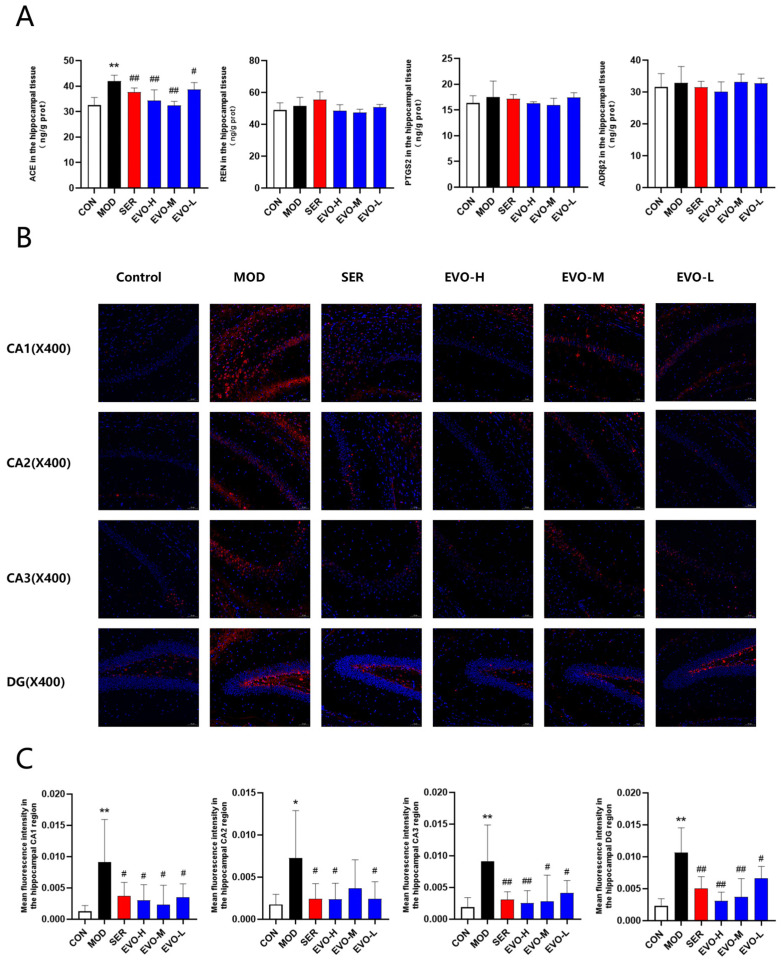
Effects of EVO on core target levels and the renin angiotensin pathway in the hippocampus of SPS-induced PTSD mice. (**A**) Evaluation of ACE, REN, ADRB2, and PTGS2 expression in the hippocampus of mice across each group. (**B**) Immunofluorescence staining of ACE in the CA1, CA2, CA3, and DG regions of the hippocampus in each group of mice; magnification ×400 (scale bar: 50 μm). Immunofluorescence staining of mouse brain slices in the hippocampal region of mice stained blue and red for ACE expression. (**C**) Mean optical densities of the CA1, CA2, CA3, and DG regions in the hippocampus of mice in each group. Data are presented as means ± standard deviations (*n* = 3). Compared with the CON group, * *p* < 0.05, ** *p* < 0.01; compared with the SPS group, # *p* < 0.05, ## *p* < 0.01.

**Figure 9 nutrients-16-01957-f009:**
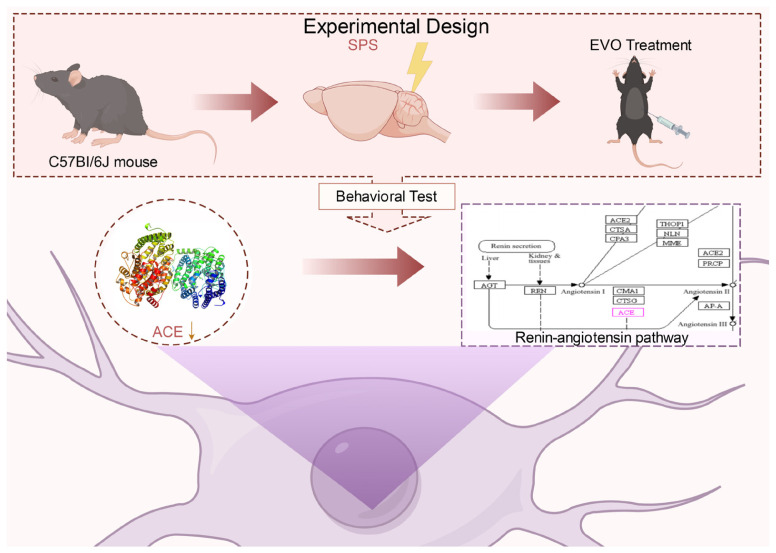
Mechanisms of evodiamine intervention in behavioral and hippocampal neurons in a mouse model of post-traumatic stress disorder.

## Data Availability

The data presented in this study are available on request from the corresponding author.

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
