# Peer review of "Effects of Evodiamine on Behavior and Hippocampal Neurons through Inhibition of Angiotensin-Converting Enzyme and Modulation of the Renin Angiotensin Pathway in a Mouse Model of Post-Traumatic Stress Disorder"

_nutrients, 2024, doi:10.3390/nu16121957_

Round 1
Reviewer 1 Report
Comments and Suggestions for Authors
The manuscript "Effects of Evodiamine on Behavior and Hippocampal Neurons through Inhibition of Angiotensin-Converting Enzyme and Modulation of the Renin-Angiotensin Pathway in a Mouse Model of Post-Traumatic Stress Disorder" delves into characterization of the histological and pharmacological changes in the hippocampus in a post-traumatic stress (PTSD) disorder mosue model as well as the behavioural changes in PTSD. The study reports the anxiolytic and antidepressant effects of EVO and subsequent memóry enhancement and cognition in behavioural tasks after SPS. Moreover, the resulty oft he study illustrate the protective effect of EVO on hippocampal neurons and the attenuation of hippocampal neuron apoptisis in SPS mice. The renin-angiotensin system (RAS), along with other canonical systems, has been reported to show important role in stress including in PTSD. The sumbitted manuscript further illucudates the role of ACE and RAS in stress PTSD and a possible thereupatic intervension in Stress and stress – related disorders.
After going through the manuscript, I have following comments for the authors.
1. Various pre-clinical studies on animal models have established a fact that the symptoms induced on the animals might not fully mimic the complexity and progression of the symptoms in the humans. Hence the PTSD induced in the Mice in this study might not completely compliment that in the humans due to various etiological factors. Please discuss this point in the manuscript for clarity in interpretating the results.
2. Please briefly mention the aim of the study in the concluding sentence of the introduction section.
3. Many abbreviations are used in the manuscript and the full form of some oft he abbreviations are missing. Please double check it.
4. Please mention the limitations of the study.
Comments on the Quality of English LanguageMinor grammatical corrections and syntax adjustments needed.
Reviewer 2 Report
Comments and Suggestions for Authors
The article “Effects of Evodiamine on Behavior and Hippocampal Neurons through Inhibition of Angiotensin-Converting Enzyme and Modulation of the Renin-Angiotensin Pathway in a Mouse Model of Post-Traumatic Stress Disorder” submitted to the MDPI journal nutrients assesses the ability of a plant derived compound evodiamine in a model of PTSD. Unfortunately, the article has significant flaws and needs to be reviewed comprehensively before being resubmitted to a journal for publication. This manuscript appears that is has not undergone appropriate revision before submission and should dbe rejected on the following grounds.
Major Comments:
1. The study is based on an “SPS-exposed” animal model, but the only reference to this model is in the methods, (citing an article that does not use the abbreviation SPS), and does not make any clarification of the abbreviation until the discussion.
2. The use of abbreviations through the manuscript is inconsistent and means the study cannot be logically followed:
a. SER is never explained – and this is an essential control. Is it positive or negative?
b. CA and DG abbreviations are not clarified – and these are the important brain regions investigated
3. The first results section (Figure 1F, G) on page 2, lines 82-85 describes “Moreover, in the elevated plus-maze test (EPMT), compared with SPS, EVO and SER led to significant increases in distance traveled in the open arms, time spent in the open arms, time spent in the open arms, and time spent in the open arms.” This shows that the wording has not been closely checked or edited.
4. Page 3, lines 95-103 does not refer to Figure 1H (latency to fall). The next two graphs (spatial exploration) should be changed to Fig 1I, and current Fig 1I should be changed to Fig 1J for clarity.
5. The Figures all describe EVO-H, M and L yet there is no explanation for these abbreviations anywhere.
6. Figure 2 legend (page 6, lines 140-141) describe regions highlighted by green boxes and orange boxes – yet there are no boxes of this colour in the figure.
7. The dose of SER control is different to all EVO doses tested, but there is no justification for this.
8. There is no description of how the TST, EPMT, or FST were analysed.
Comments on the Quality of English LanguageThe English could benefit from a review, however it is hard to adequately assess the language due to inconsistencies in abbreviations, and poor structure.
Reviewer 3 Report
Comments and Suggestions for Authors
In this manuscript the authors investigate the effect of EVO treatment on (PTSD) post traumatic stress
disorder model-induced changes in behaviour and hippocampal neurons. In addition, they explore the
mechanisms of EVO intervention in PTSD trough Network pharmacology, non targeted serum metabolomics
techniques and intersecting KEGG pathways. The authors demonstrate that EVO treatment is able to mitigate
sps induced memory and cognitive deficits and is able to modulate ACE levels significantly increased in PTSD
mice model. Probably regulating hippocampal ACE levels EVO can modulate the renin-angiotensin pathway
and exert a therapeutic effect on PTSD.
Same points that the authors should reviewed:
Major concern
1) The weakest aspect of the paper is the effect of the PTSD mice model and EVO modulation in hippocampus.
In detail: results clearly demonstrate the ability of EVO to mitigate SPS induced memory and cognitive deficits
as well as the increased anxiety and depression. However, SPS-induced alterations in hippocampus
morphology and its ability to induce neurodegeneration are not so convincing. It is very hard to find
differences between the control and the sps model as well as between the model and EVO treatment. By the
title and the abstract (neuroprotective effect of EVO on the hippocampus of SPS mice (line 26)) a significant
effect of the treatment on hippocampal neurons would be expected, but that appears not to be the case,
because the effect of the model is not so evident. I suggest to give less emphasis to this aspect by replacing
some words such as notable (pag 6; line 148). In addiction morphological changes, signs of
neurodegeneration (as reported in the conclusion) as well as less pigmentated Nissl vesicles, should be clearly
pointed in the figures to better evaluate the effect of the model and of EVO treatment.
2) OFT data are not clear:
- Pag 2; line 81: Evo and SER treatment don’t increase the centroid latency time but on the contrary, reduce
it as showed in the second panel fig 1B. In addition, in this part of the text centroid latency time is reported
while in fig1B the authors write incubation period in the centre. Are both of them the time in the centre?
Please explain.
- Pag 2, line 85: I don’t agree with this sentence. To demonstrate that SPS model is able to hinder locomotor
activity, the total distance travelled and the speed in the OFT should be showed too.
In addition, can the author explain the apparent discrepancy between the reduction of the distance travelled
in the centre and the increase of time in the centre of SPS mice?
Pag 17-18; lines 342-345: this part should be reviewed, SPS stimulation doesn’t modify the expression
levels of the core targets REN, ADRB2, and PTGS2, the only change is the increase od ACE levels.
3) Discussion, in general it appears lacking because there are many interesting data that are not discussed
in the paper, below some suggestions:
- what about hippocampal neurodegeneration in other PTSD animal models? PTSD model-induced changes
in hippocampal morphology and vitality should also be discussed with other animal models not only respect
to patients, to render the information more solid.
- in the results the authors highlight the effect of SPS model and EVO administration on neuroaminic acid
(pag 12 line 240), which is the meaning of this modulation? A discussion on this aspect should be included.
The authors should explain because some words in the figures (fig 9 for example), are highlighted in red. Do
these words represent crucial mechanisms through which EVO act? This aspect should be discussed.
Minor concerns
4) Based on the consideration on hippocampal neurodegeneration (point 1)) the title should be reviewed.
5) Pag 2; line 68-69: a reference for EVO antidepressant like activity and its ability to improve cognition
should be included.
6) Pag 2; line 91 ACE treatment should be corrected with EVO treatment
7) Fig 1 B consists of 2 graphs as well as figure F (2 graphs) and Fig H (3 graphs). All graphs should be
numbered with a corresponding letter.
8) Legend fig 1, the letter (H) is missing.
9) Pag 10; line 234, which is the upregulated metabolite? The figure 1 C is illegible.
10) Attention for the acronyms used in the text, all the abbreviations should be explained.
11) Pag 2; line 84 there is a repetition in the text
12) the quality of fig 5 and 6 should be improved, you can’t read anything
Round 2
Reviewer 3 Report
Comments and Suggestions for Authors
The suggestions are included in the attached file
